# Sentiment and Moral Narratives during COVID-19

**Oscar Araque**[1*]**, Kyriaki Kalimeri**[2*]**, Lorenzo Gatti**[3]

[1] Intelligent Systems Group, Universidad Politécnica de Madrid, Madrid, Spain
[2] Data Science & Digital Philanthropies Laboratory, ISI Foundation, Turin, Italy
[3] Human Media Interaction Lab, University of Twente, Enschede, The Netherlands
`o.araque@upm`, `kkalimeri@acm.org`, `l.gatti@utwente.nl`
[*] These authors contributed equally to this work.

## Abstract

The COVID-19 pandemic radically changed the lives of millions with unprecedented economic, societal, but also psychological consequences. Containing the pandemic urged for rapid measures, the success of which hugely rely on mass cooperation. Here, we employed a sentiment (DepechMood++) and a morality (MoralStrenght) dictionary, to analyse user-generated text on the Twitter platform. Quantifying cooperation and competitive attitudes from the linguistic cues during the pandemic, we depict the moral profiles associated with each position. Moreover, we discuss the moral and emotional profiles of individuals who tweeted about conspiracy theories, and the evolution of moral values throughout the pandemic period for four case-study countries. Our findings show the evolution of moral values during the pandemic for Italy, United States, New Zealand and China. Comparing the moral profiles of cooperative versus competitive attitudes, we find that high loyalty and consideration of the group members is positively correlated to the sentiment of inspiration and negatively correlated to fear.

## 1 Introduction

The novel coronavirus (COVID-19) is a massive global health crisis calling for rapid measures to contain it. These require large-scale behaviour changes which place a significant psychological burden on individuals. As expected, in times of uncertainty, disinformation, panic messaging, and conspiracy theories proliferate online. Conspiratorial thinking, in particular, has many troubling consequences such as antisocial behaviour, rejection of science, decreased trust in government, and a lack of civic engagement.

Moral norms and values regulate the behaviour of individuals living in communities. Social media platforms empower all to express their beliefs and personal narratives. However, at the same time, these platforms allow for rapid spreading of disinformation or panic messages. Here, we attempt to understand citizens' beliefs and attitudes towards the pandemic, analysing the moral narratives of user-generated content on the Twitter platform. The moral rhetoric inferred from the tweets, provide insights on how the public at large is responding to the outbreak.

In particular, we focus our analysis around two major research questions:

**RQ1: Are there predominant moral values expressed by those who proliferate conspiracy theories?** People supporting conspiracy theories have similarities in their moral narratives and worldviews. We provide insights on the moral profiles associated with well-known conspiracy theories around the pandemic (e.g. 5G antennas).

**RQ2: Do people cooperate rather than compete in response to a crisis?** Instinctual survival behaviours are entwined in human nature and compete with notions of altruism. We compare how moral rhetoric evolved during the pandemic with the in four exemplar countries.

Quantifying the moral narratives via user-generated text, we sketch the moral profiles of individuals susceptible to disinformation as well as the characteristics of the most collaborative populations. Despite the preliminary nature of our study, we show proof of the concept that our worldviews act as a lens to the information we are exposed to. Moreover, sentiment analysis on the same populations shows that collaborative behavioural attitudes

are associated with positive emotions, rather than fear.

Our findings indicate that real-time monitoring of sentiment and moral values expression can help policymakers design proactive communications campaigns and anticipate emerging social threats.

## 2 Data Collection

In this study, we employ data from the Coronavirus Twitter Data collection provided by Huang et al. (2020). The original dataset contains Twitter IDs, which were used to download the original data directly from Twitter. Additionally, IDs are accompanied by the date, keywords related to COVID-19, and the inferred geolocation. For the sake of computational efficiency, we randomly sampled 5 million tweet IDs out of the original Coronavirus Twitter Dataset.

Out of the 5 million tweet IDs, 4,380,000 were still existing and hence successfully recovered using the Twitter API[1]. Retaining only the tweets in the English language, we derived to a total number of tweets 3,163,500. All tweets are associated with the location given in the original dataset.

## 3 Materials and Methods

We operationalise morality via the Moral Foundations Theory (MFT) (Graham et al., 2009), which expresses the psychological basis of morality in terms of innate intuitions, defining the following five foundations: *care/harm*, *fairness/cheating*, *loyalty/betrayal*, *authority/subversion*, and *purity/degradation* (see (Haidt and Graham, 2007; Haidt and Joseph, 2004)). These dimensions can be grouped into two hyper foundations, namely social binding, and individualism. Even if in its infancy, MFT is the most well-established theory in the psychology and social sciences.

### 3.1 Moral and Sentiment Lexicons

To assess the moral rhetoric of the user-generated content, we were based on the *MoralStrength* lexicon proposed by Araque et al. (2019a). This dictionary is a unique resource covering approximately 1,000 lemmas combined with a crowdsourced numeric assessment of their *Moral Valence*. Moral valence indicates the strength with which a lemma

is expressing the specific moral. Each lemma is rated in a linear scale from 1 to 9, being 5 the 'neutral' value, where 9 indicates the virtue (Care, Fairness, Loyalty, Authority, Purity) and 1 the vice (Harm, Cheating, Betrayal, Subversion, Degradation) of each foundation respectively.

Despite this being a unique resource, the number of lemmas is limited. For this reason, we expanded the *MoralStrength* dictionary with word embeddings, following the methodology proposed for sentiment analysis by Araque et al. (2019b). For each word of the original lexicon, we find the 10 closer neighbouring words, as computed by a pretrained embedding model presented by Bojanowski et al. (2017). To avoid inclusion of very similar variations of a word (e.g., *cat* and *cats*), we perform both lemmatisation and filter by means of the Levenshtein distance. Also, neighbour words that have a similarity lower than 0.5 (as computed by the embedding model) are discarded to avoid too dissimilar terms. The moral valence of the word is set equal to the original term. After this expansion process, we obtain a lexicon with more than 3,000 lemmas, which augments the coverage of the resource.

Aside from the moral analysis, we also perform sentiment analysis employing the DepecheMood++ lexicon (Araque et al., 2019b). This resource offers a finer-grained emotion analysis with respect to other resources. Namely, fear, amusement, anger, annoyance, indifference, happiness, sadness, emotions are included, essential for getting a holistic understanding of the users' intend.

### 3.2 Moral and Sentiment Assessment

We employ the lexicons described above to predict both the moral valence and the sentiment of each tweet. The methodology used is straightforward; for each word in the tweet, we obtain the annotations from both lexicons, if any. Then, for each tweet, we average the obtained annotations over the eight sentiments and five moral dimensions.

## 4 Results and Discussion

Quantifying the cooperation and competitive attitudes during the pandemic is certainly not an easy task. To get a grasp of peoples' attitudes, we created two major behavioural groups. This classification was inspired by the work of (Shanthakumar

---

[1] https://developer.twitter.com/

et al., 2020); the first group focuses on collaborative nature hashtags such as socialdistancing, while the second one covers the most common conspiracy theories around this pandemic. Table 1 reports the terms we employed to create the two groups, which may appear as hashtags or keywords. We also report the total number of tweets for each hashtag or keyword.

| Group | Hashtag | Appearances |
|---|---|---|
| SD | #StayAtHome | 3,943 |
| | #SocialDistancing | 3,309 |
| | #StaySafe | 4,199 |
| | #StayHome | 8,588 |
| Total | | 20,039 |
| M | #Spy | 1,094 |
| | #ChinaVirus | 739 |
| | #Trump | 249,989 |
| | #ObamaGate | 913 |
| | #5G | 554 |
| | #BillGates | 596 |
| Total | | 253,885 |

Table 1: Number of tweets per hashtag for the Social Distancing (SD) group and the Misinformation propagators (M) group, inspired by (Shanthakumar et al., 2020).

**Are there predominant moral values expressed by those who proliferate conspiracy theories?** In times of uncertainty and distress, people often reason instinctually. Understanding in-depth the human values triggered and who may resonate with those, is essential to fight the info-demic. Table 2 reports the morality scores of our two groups. We notice an overall homogeneous moral profile between the two groups, with major characteristics being the high levels of care, loyalty, and purity for the SD group. Notions of empathy, compassion, and protection of others are the core values of this group. Interestingly, loyalty value is solid too. Loyalty is related to the obligations in the group membership, including self-sacrifice. Purity is also significantly higher in this group, expressing concerns about physical contagion. Finally, this group is more authoritative with a pronounced feeling related to the obligations of social relationships (see (Haidt and Graham, 2007; Haidt and Joseph, 2004)).

On the other hand, mis- and disinformation messaging seem to reflect more on individuals who represent the exact opposite values. Being low

in care, loyalty, purity and authority, this group is not mainly concerned with the physical contagion. Intergroup competition is far more decisive for survival, leading the individuals in this group to downplay the severity of the virus and defy the measures.

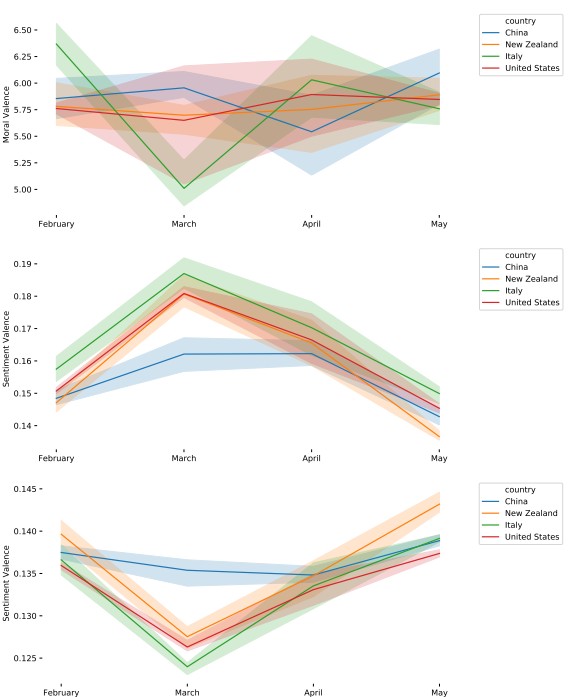

Figure 1: Evolution of Loyalty (upper) and Fear (center) and Inspiration (lower) in tweets during the pandemic in several case study countries.

**Do people cooperate rather than compete in response to a crisis?** The loyalty foundation expresses cooperation or competition. Individuals who believe in their group promote self-sacrifice and cooperation. Visualising the evolution of loyalty during the pandemic in Figure 1, we notice that there is a quite big fluctuation in values during the time. Importantly, the time trends are different for different countries; this may be due to several reasons. Correlating[2] the moral values with the emotions obtained by each tweet, we find the strongest negative correlation with fear (-.22) and the strongest positive ones with inspired (.33). Without implying causation, this indicates that collaborative attitudes are related to positivity and inspiration rather than doctrines of fear.

---

[2]With the term "correlation" we refer to the Pearson's correlation coefficient.

| Keyword | C/H | F/C | L/B | A/S | P/D | Fear (%) |
|---|---|---|---|---|---|---|
| #StayAtHome | 6.07 | 7.51 | 6.39 | 6.15 | 6.58 | 26 |
| #SocialDistancing | 6.66 | 7.60 | 6.37 | 6.05 | 5.44 | 24 |
| #StaySafe | 6.94 | 6.99 | 7.16 | 6.39 | 6.10 | 29 |
| #StayHome | 6.32 | 7.11 | 7.08 | 6.52 | 5.64 | 30 |
| **SD average** | 6.47 | 7.27 | 6.85 | 6.34 | 5.60 | 28 |
| #Spy | 3.56 | 6.96 | 2.99 | 7.15 | 3.44 | 34 |
| #ChinaVirus | 3.45 | 7.20 | 4.48 | 5.86 | 2.82 | 45 |
| #Trump | 4.37 | 6.72 | 5.42 | 5.85 | 4.12 | 45 |
| #ObamaGate | 3.96 | 7.76 | 3.19 | 5.19 | 4.21 | 12 |
| #5G | 3.93 | 6.70 | 4.91 | 5.19 | 3.22 | 39 |
| #BillGates | 4.24 | 7.78 | 6.22 | 6.82 | 3.79 | 40 |
| **M average** | 4.37 | 6.73 | 5.31 | 5.85 | 4.11 | 45 |

Table 2: Averaged moral values organized by hashtag. Care/Harm (C/H), Fairness/Cheating (F/C), Loyalty/Betrayal (L/B), Authority/Subversion (A/S), and Purity/Degradation (P/D). Higher scores are related to the first term of the moral dimension, while lower scores with the second.

## 5 Conclusions

Despite the preliminary nature of our study, our approach opens new research avenues to mitigate the impact of the infodemic.

Quantifying the moral rhetoric and emotional triggers, we can derive a holistic view of the individual and identify the most vulnerable populations. Such insights may also provide evidence-based responses to policymakers on societal distress, or moral polarisation, contributing to anticipate threats.

Our descriptive findings show that there are essential variations in the reactions of individuals across countries but also throughout the various pandemic phases. The main sentiment of the pandemic was fear, accounting for the 48% of the emotions expressed in our data. The 'fear' category may include economic distress, loneliness, or even health concerns, which justifies the high frequency in tweets.

A straight forward step would be to combine community detection algorithms with moral narrative recognition, to derive the moral and sentiment profiles of the spreaders of disinformation as well as those of the most susceptible individuals. This direct future research line can help policymakers to act timely and devise better policies.

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
