# OpenReview forum: "Sentiment and Moral Narratives during COVID-19"
_EMNLP/2020/Workshop/NLP-COVID — Submitted to NLP-COVID19-EMNLP_

### Official Review · AnonReviewer2 · 2020-09-16
**Sentiment and Moral Narratives during COVID-19**

**Rating:** 5
**Confidence:** 4

**Review:**

## SUMMARY

This work characterized the sentiment and moral analysis based on more than 3M English Tweets from the Coranavirum Twitter Data.  The data analysis was performed according to two lexicons: 1) MoralStrength lexicon for moral valence based on Moral Foundation Theory (MTF),  2) DepecheMood++ for sentiment scale calculation. Moral lexicons were expanded based on word embeddings. Each word in the tweet was annotated based on these two lexicons and each tweet is marked by the average value over 8 sentiments and 5 moral dimensions.  Tweets frequencies were then calculated for two behavioral groups: Social Distancing(SD) group and the Misinformation propagators(M) group.  Average morality values were further compared for the two groups.  Loyalty (Morality), fear (Sentiment), and inspiration (Sentiment) evolutions were studied from February to May based on 4 different countries: China, New Zealand, Italy, US.

In this reviewer's view, this work is a worthwhile contribution. It provides special insights into the relationships between behavior and morality and sentiment for the COVID dataset.

## COMMENTS

1. Please clarify how the annotation is performed. Is it to match each word to the lexicons in the two dictionaries(morality and sentiment)?

2. The 'Hashtag/keywords' in Table 1 and 2, are they listed based on certain criteria or just the keywords related to COVID19 from the original Twitter data?

3. 'Fear' is a sentiment category. What does the 'Fear' column mean in Table 2 (the averaged moral values)? Is it the fear score or the ratio of posts?

4. In Figure 1, for each line, there is a shaded area. Is it the range of scores?

6. From Table 1, the sum of the number of posts for M and SD groups is far less than 3,163,500 (the total number of tweets being collected). Please clarify after grouping to M and SD, where do the rest of the posts go?

---

### Official Review · AnonReviewer3 · 2020-09-21
**Interesting methods used to analyse morality and sentiments in Twitter, however the explanations and results are not  clear/strong enough.**

**Rating:** 6
**Confidence:** 3

**Review:**

The work sets out to answer two COVID related questions looking into Twitter data. The questions are if people cooperate or compete at the time of crisis and what are the major moral values of people believing in conspiracy theories. Each tweet is annotated using a moral and a sentiment lexicon. Also based on presence of certain hashtags, all the tweets are divided intto two categories. The research questions are answered based on the average moral and sentiment score of each tweet.

Comments:

The paper needs to be proofread. There are sentence structure issues throughout the paper. Some examples: “with the in four exemplar countries” , “communications Campaigns”, “derived to a total number Of”, “we were based on”, “being 5 the”, “understanding of the users’ intend”, “is a quite big fluctuation”.

The choice of hashtags for breaking the tweets to two behavioural groups needs to be explained.

It could be useful to give some examples of how the scoring is done.

Need to be clarified if only 273,924 out of 3,163,500 tweets were used for the analysis. If so this need to be explained in the data collection section.

Figure 1 the y-axis could be labelled with (loyalty,fear and inspiration).

The answer to question two is based on the negative correlation of fear and loyalty but Loyalty does not seem to follow the fear trend. Even when fear goes down loyalty does not keep going up. Given we cannot explain the fluctuations in the loyalty the answer to question 2 is not satisfactory.

---

### Official Review · AnonReviewer1 · 2020-09-25
**Interesting idea, but needs further development**

**Rating:** 4
**Confidence:** 3

**Review:**

# [REVIEW] COVID-19/EMNLP Workshop -- Sentiment and Moral Narratives during COVID-19 [paper 18]

10th Sep 2020

## SUMMARY

This short paper presents a lexicon-based approach to understanding moral narratives in English-language Twitter data.  It’s an interesting approach, and nice to see a paper that utilises psychological theory.  The paper presents 2 core results:

1. Tweets pertaining to social distancing are characterised by empathy, compassion, and protection, and tweets pertaining to misinformation characterised by the opposite qualities
2. Tweets pertaining to loyalty fluctuate over time

My overall view is that this paper and line of research is promising, but currently needs a little more time to marinate before it is ready for publication.


## COMMENTS

* The writing is generally comprehensible, but there are some points that would benefit from being tightened up.  Examples include:
 “ Containing the pandemic urged for rapid measures, the success of which hugely rely on mass cooperation” and“Understanding indepth the human values triggered and who may resonate with those, is essential to fight the infodemic.”
* Suggest listing 4 exemplar countries on first page in the body of the paper (although they are currently listed in the abstract)
* The use of embeddings to expand the lexicon seems reasonable, but was there an attempt at evaluating the words identified?
* Technical details are somewhat lacking (e.g. packages used for word embedding work)
* My understanding is that MFT, while very influential in moral psychology, is also quite controversial and competes with stage based models and theories that are more influenced by moral philosophy (e.g. Joshua Greene’s work).  However, I think the choice of MFT is sensible given your goals, but could be expanded on a little.  Also, it would be helpful to explicitly state that the MoralStrength lexicon explicitly draws from MFT
* Minor point, but in Section 4, \citet{} should probably be used to reference the Shanthakumar paper.
* “indifference, happiness, sadness, emotions are included, essential for getting a holistic understanding of the users’ intend.”  gaining a holistic understanding?
* Table 1.  When you say that the choice of hashtags was inspired by Shanthakumar, did you make any changes?
* “Finally, this group is more authoritative with a pronounced feeling related to the obligations of social relationship” I’m not sure if you mean authoritarian here?
* I’m not clear on how reliable the Grouping of hashtags is.  A quick search of twitter suggests that the hashtag #Trump is used both to propagate misinformation, and to attempt to correct it.
* Figure 1 — text is too small to read - also, I’m not sure how useful the confidence intervals are
* Figure 1 — emotions (i.e. loyalty, etc) should probably be presented on the graph, not just in the caption
* I’m a bit confused about why you chose the countries you did given that all your tweets were in English.  How representative are English language tweets generated from Italy? Also, isn’t Twitter severely restricted in China?
The choice of categories chosen — loyalty, fear, inspiration — don’t seem to map very well to MFT
* Evaluation is currently somewhat lacking